# Research on the Multimode Switching Control of Intelligent Suspension Based on Binocular Distance Recognition

**Chen Huang [1,2], Kunyan Lv [1,*], Qing Xu [2] and Yifan Dai [2]**

1 Institute of Automotive Engineering, Jiangsu University, Zhenjiang 212013, China; huangchen@ujs.edu.cn
2 State Key Laboratory of Automotive Safety and Energy, Tsinghua University, Beijing 100084, China; qingxu@tsinghua.edu.cn (Q.X.); df08@mails.tsinghua.edu.cn (Y.D.)
* Correspondence: 17712603097@163.com

**Abstract:** As the upgrade of people's requirements for automotive driving comfort, conventional passive suspensions for cars have fallen short of existing demands due to their nonadjustable damping and stiffness, so semiactive suspensions and active suspensions have gained growing acceptance. Compared with active suspensions, semiactive suspensions offer the advantages of a low manufacturing cost and reliable structure, and thus have become the preferred choice for most vehicles. To optimize the control effect of semiactive suspensions under different working conditions, this paper completed the modeling of magnetorheological semiactive suspension system dynamics and road inputs; then, the design of binocular camera sensing algorithms was performed to obtain the real-time distance of the target using the point cloud ranging function, and the parameters required for suspension control were also obtained. This was followed by the completion of the control-mode-switching rules and the design of the suspension controller. According to the different control objectives, the mode could be divided into the obstacle-road mode, straight-road mode, and curved-road mode. The suspension controller included the BP-PID (neural network PID controller) controller and the force distributor. Finally, the effectiveness of the mode-switching rules and the control method was verified through system simulation and the hardware-in-the-loop test.

**Keywords:** semiactive suspension; vehicle control strategy; target recognition; BP neural network



## 1. Introduction

As major component of the automotive chassis system, the suspension directly affects the ride comfort of a vehicle [1]. Conventional passive suspensions have not been able meet the existing demand because they cannot change the damping and stiffness. Active suspensions have excellent performance but cannot be put into large-scale application. In contrast, semiactive suspensions are less costly and have an adjustable damping force. By applying control algorithms to semiactive suspensions, they can be adaptively adjusted according to the current and future road conditions, which is a better choice for suspension systems currently [2–5].

As a widely used senor in vehicles, the vehicle camera has a significant contribution in vehicle control and assisted driving. As a sensor built on the principle of binocular stereo vision, the binocular camera has both image-acquisition and range-measurement functions. This camera captures images from different angles simultaneously using a binocular stereo vision system consisting of two cameras, which is then processed by a matching algorithm to obtain a parallax map. Finally, it calculates the distance information of the object to be measured by combining the camera calibration parameters. A simple binocular system can be completed using two on-board monocular cameras, which has great application prospects [6–9].

For semiactive suspensions, many studies have been conducted at home and abroad. In terms of the suspension structure, Hu et al. proposed a magnetorheological damper

with an improved structure. They increased the magnetic field range by connecting the radial gap and axial gap, which improved the response speed of the damper and increased the adjustment range [10]. Brooks in the UK designed a current–variable liquid damper using the current–variable effect to control the flexible diaphragm, resulting in a smoother and more homogeneous process of the damping force change [11]. In terms of suspension control, there are currently the PID, LQG, MPC, sliding film control methods, and so on [12–16]. Neural networks for the online tuning of PID parameters have been widely used because of their flexibility and relatively mature technology [17–20]. For example, Li Mei et al. proposed a fuzzy neural network parameter optimization algorithm combining the particle swarm optimization algorithm and the gradient descent method. They proposed to adjust the parameters of the active suspension PID controller in real time with body acceleration as the main optimization objective [1]. However, prescanning control, which can be adjusted in real time according to the road conditions ahead, has gradually gained popularity among researchers in recent years. Bender first proposed the concept of prescanning control and applied it to the control of a single-degree-of-freedom suspension system [21]. R.S. Sharp et al. applied wheelbase prescanning to the design of an active suspension. This system relied on the prescanning information provided by the front wheels to apply control to the rear wheels, and the results indicated that wheelbase prescanning significantly improved the performance of the rear suspension [22]. C. Gohrle proposed two model-predictive approaches for the preview of active suspension controllers and compared them with the well-known optimal-preview control approach. The first model-predictive controller optimized the actuator displacements on a nonequidistant grid over the preview horizon. The second controller optimized the trajectories for the heave, pitch, and roll of the vehicle body over the preview horizon using a quadratic program [23]. Panshuo Li et al. from the University of Hong Kong proposed a multiobjective control method for active suspension based on wheelbase prescanning. They compared the active suspension performance under different operating conditions and found that the body droop and angular accelerations were significantly improved as compared to the control method without wheelbase prescanning [24].

However, most of the current semiactive suspension control methods do not consider the main factors affecting ride comfort under different working conditions. This results in the fact that a single control mode cannot be adapted to all applications. Therefore, in this paper, a binocular camera was adopted to complete target recognition and prescanning information extraction. That is, the speed bumps and the start points of the lane-line circle curve were used as the judgment rule for the suspension mode so as to optimize the body posture under different scenarios and improve the ride comfort and handling stability.

The rest of this paper is organized as follows. System modeling is described in Section 2, including the seven-degrees-of-freedom (DoF) full-vehicle model, the road-surface input model, and the mathematical model of magnetorheological dampers. A target-distance-recognition method based on the binocular camera is described in Section 3. In Section 4, an intelligent-suspension control system is presented. In Section 5, an HIL experiment is performed to illustrate the effectiveness and advantages of the proposed scheme. Finally, the work of the full paper is summarized and the validity of the proposed program is clarified in Section 6.

## 2. System Modeling

### 2.1. Vehicle Dynamics Modeling

The seven-degrees-of-freedom whole-vehicle dynamics model is shown in Figure 1. The model is more consistent with the actual vehicle dynamics, so this paper chooses to use this model to develop the control algorithm [2]. It includes the body roll, body pitch, and body droop, as well as the droop movement of all four wheels. Where the subscripts n represent the left front wheel, right front wheel, left rear wheel, and right rear wheel, respectively; $f_n$ is the controlled damping force; $z$ refers to the vertical displacement of the body centroid; $\theta$ represents the body pitch angle; $\phi$ is the body roll angle; $z_{sn}$ is the

vertical displacement of the body at the corresponding position; $z_{un}$ signifies the vertical displacement of the corresponding wheel; $m$ is the upper spring mass; $m_{un}$ is the lower spring mass at each position; a, b, c, and d, respectively, denote the distance from the center of vehicle to the front, rear, left, and right ends; $k_{tn}$ is the stiffness of each tire; $k_1$ and $k_2$ are the stiffness of the front suspension elastic element; k$_3$ and k$_4$ refer to the stiffness of the rear suspension elastic element; $c_n$ is the passive suspension damping force; $I_y$ is the pitch inertia, while $I_x$ denotes the lateral tilt inertia; $q_n$ signifies the road input.

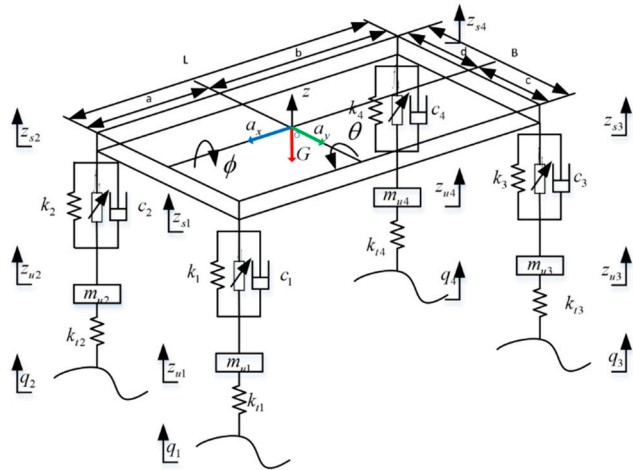

**Figure 1.** Vehicle dynamics model.

The vehicle dynamics equations include:

$$m\ddot{z} - [k_1(z_{u1} - z_{s1}) + c_1(\dot{z}_{u1} - \dot{z}_{s1}) + k_2(z_{u2} - z_{s2}) + c_2(\dot{z}_{u2} - \dot{z}_{s2}) + k_3(z_{u3} - z_{s3})$$
$$+ c_3(\dot{z}_{u3} - \dot{z}_{s3}) + k_4(z_{u4} - z_{s4}) + c_4(\dot{z}_{u4} - \dot{z}_{s4})] +$$
$$f_1 + f_2 + f_3 + f_4 = 0 \tag{1}$$

$$I_y\ddot{\theta} - [k_1(z_{u1} - z_{s1}) + c_1(\dot{z}_{u1} - \dot{z}_{s1}) + k_2(z_{u2} - z_{s2}) + c_2(\dot{z}_{u2} - \dot{z}_{s2}) - f_1 - f_2]a +$$
$$[k_3(z_{u3} - z_{s3}) + c_3(\dot{z}_{u3} - \dot{z}_{s3}) + k_4(z_{u4} - z_{s4}) + c_4(\dot{z}_{u4} - \dot{z}_{s4}) - f_3 - f_4]b = 0 \tag{2}$$

$$I_x\ddot{\phi} - [k_2(z_{u2} - z_{s2}) + c_2(\dot{z}_{u2} - \dot{z}_{s2}) + k_4(z_{u4} - z_{s4}) + c_4(\dot{z}_{u4} - \dot{z}_{s4}) - f_2 - f_4]c +$$
$$[k_1(z_{u1} - z_{s1}) + c_1(\dot{z}_{u1} - \dot{z}_{s1}) + k_3(z_{u3} - z_{s3}) + c_3(\dot{z}_{u3} - \dot{z}_{s3}) - f_1 - f_3]d = 0 \tag{3}$$

$$m_{u1}\ddot{z}_1 - k_{t1}(q_1 - z_{u1}) + k_1(z_{u1} - z_{s1}) + c_1(\dot{z}_{u1} - \dot{z}_{s1}) - f_1 = 0 \tag{4}$$

$$m_{u2}\ddot{z}_2 - k_{t2}(q_2 - z_{u2}) + k_2(z_{u2} - z_{s2}) + c_2(\dot{z}_{u2} - \dot{z}_{s2}) - f_2 = 0 \tag{5}$$

$$m_{u3}\ddot{z}_3 - k_{t3}(q_3 - z_{u3}) + k_3(z_{u3} - z_{s3}) + c_3(\dot{z}_{u3} - \dot{z}_{s3}) - f_3 = 0 \tag{6}$$

$$m_{u4}\ddot{z}_4 - k_{t4}(q_4 - z_{u4}) + k_4(z_{u4} - z_{s4}) + c_4(\dot{z}_{u4} - \dot{z}_{s4}) - f_4 = 0 \tag{7}$$

*2.2. Mathematical Model of the Magnetorheological Dampers*

The magnetorheological damper mathematical model is of great importance for the calculation of the damping force of the semiactive suspension, and a good mathematical model can significantly improve the effectiveness of the control algorithm. For the consideration of computational efficiency, as well as the accuracy, a nonparametric model established by the motion parameters of the damper is widely used at present. The polynomial

model proposed by Choi [25] is used to model the damping force based on the speed of the motion of the damper, with the damping force being:

$$F_d = \sum_{i=0}^{n} a_i v^i \tag{8}$$

where $i$ is the polynomial order $i = 1, 2, 3, \ldots, n$, which is used to simulate the magnetorheological damper hysteresis characteristics and improve the model accuracy, with $n > 5$ in general; $v$ is the speed of the damper motion; $a_i$ represents the polynomial model coefficient, which is obtained by fitting to the experimental data, and its relationship with the current is:

$$msa_i = B_i + C_i I \tag{9}$$

So, the damper damping force is:

$$F_d = \sum_{i=0}^{n} (B_i + C_i I) v^i \tag{10}$$

This provides the magnitude of the control current:

$$I = \frac{F_d - \sum_{i=0}^{n} B_i v^i}{\sum_{i=0}^{n} C_i v^i} \tag{11}$$

Characterization tests were conducted on a prototype magnetorheological damper. The arrangement of the test bench was referred to as the "Test Method of Cartridge Damper Bench QC/T395-1999". The test rig included the upper computer, controller, excitation head, sensors, and other equipment. During the test, the magnetorheological damper was excited in the vertical direction to produce a simple harmonic motion with a fixed stroke in the middle of the test bench, and the test bench can be found in Figure 2. The supply current used in the test was varied in the range of 0–2 A at an interval of 0.2 A. After each current value was stabilized, cyclic loading was performed at different excitation frequencies. The average value of the recorded data was then taken as the test result to obtain the damper characteristics data under different currents. The results can be observed in Figure 3. It could be seen that, when the current was certain, the damping force was proportional to the speed of the magnetorheological damper. And, as the speed increased, the separation of the upper and lower branch curves of the magnetorheological damper demonstrated the characteristics of nonlinearity.

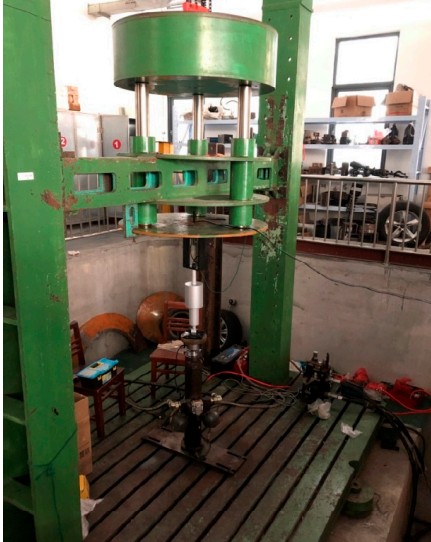

**Figure 2.** Bench test layout.

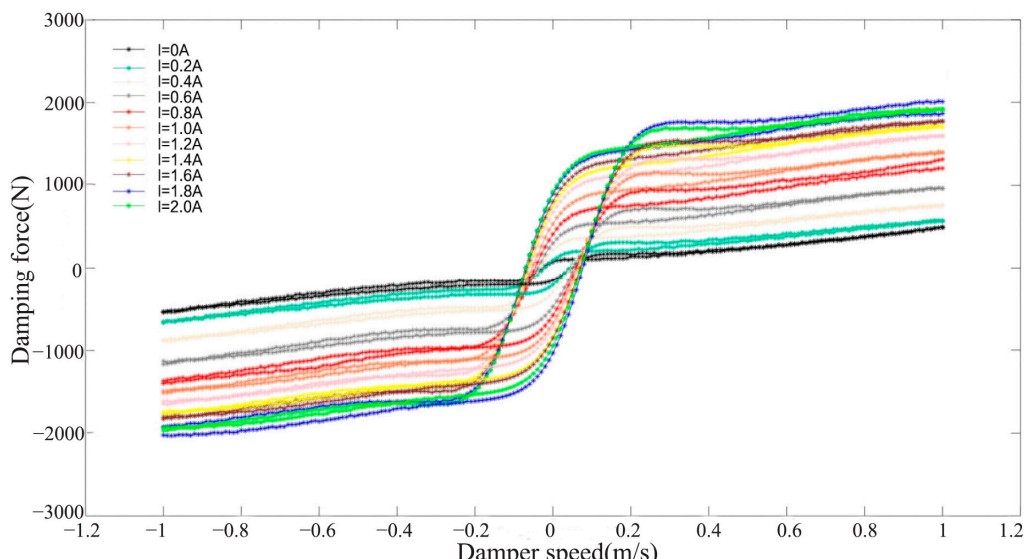

**Figure 3.** Damp characteristic curve.

The maximum current 2 A has been specified in the text. From the figure, it can be seen that the magnitude of the damping force is proportional to the speed of the magnetorheological damper when the current is certain. And, with the increase of speed, the separation of the upper and lower branch curves of the magnetorheological damper is more obvious. The simulation and experimental comparison diagrams are shown in Figure 4.

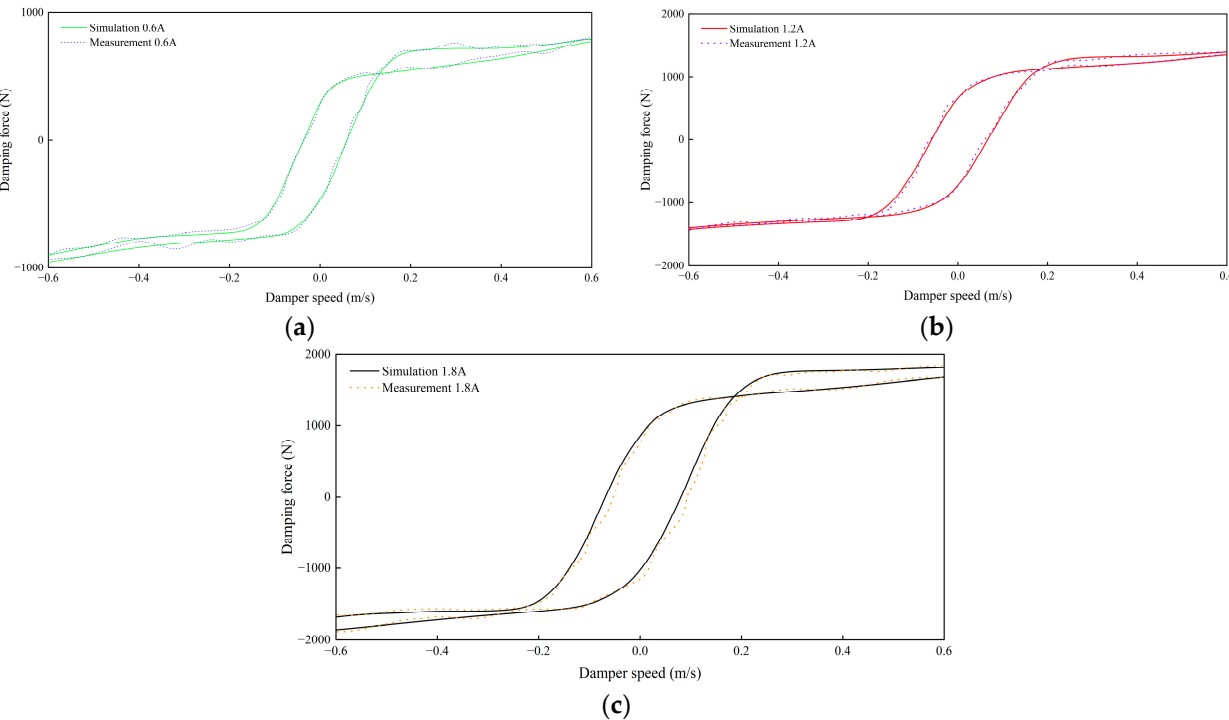

**Figure 4.** Simulation and measurement comparison diagrams: (**a**) I = 0.6 A; (**b**) I = 1.2 A; (**c**) I = 1.8 A.

### 2.3. Road-Surface Input Model

The power spectral density of the road can be expressed according to the Draft Method for Representation of Road Unevenness proposed by the International Organization for Standardization in ISO/TC108/SC2N67 [26]. Then, the spatial frequency power spectral

density was converted to the temporal frequency power spectral density to obtain the filtered white noise pavement model, and the equation is shown below:

$$\dot{q}(t) = -2\pi f_0 q(t) + 2\pi n_0 w(t)\sqrt{G_q(n_0)v} \tag{12}$$

where $w(t)$ is the Gaussian white noise and $v$ is the vehicle speed; $f_0$ refers to the lower limit as of frequency, which is generally taken as 0.01.

Since this paper is based on the whole-vehicle model for control-method development, it is necessary to build a four-wheel input model with a single-wheel pavement-unevenness input. Assuming that the vehicle travels at a uniform speed, there is a time delay between the road excitation of the front and rear wheels, which can be calculated by the following equation:

$$t = \frac{L}{v} \tag{13}$$

where $L$ is the front and rear wheelbase, and $v$ denotes the vehicle speed.

Considering the actual suspension structure and the connection of various parts of the vehicle body, there is a mutual interference relationship between the left and right wheel road-surface-unevenness input. Normally, the coherence is stronger in the low-frequency region and weaker in the high-frequency region, so a low-pass filter was applied to save the low-frequency strong-coherence region of the left wheel road-surface-unevenness input, and the high-pass filter was used to save the high-frequency low-coherence region of the random road surface unevenness; the two regions could be synthesized to generate the right wheel road-surface-unevenness input [27].

When the vehicle speed reached 10 m/s and the road surface was B-grade, the four-wheel pavement-unevenness input obtained from the simulation could be shown, as in Figure 5:

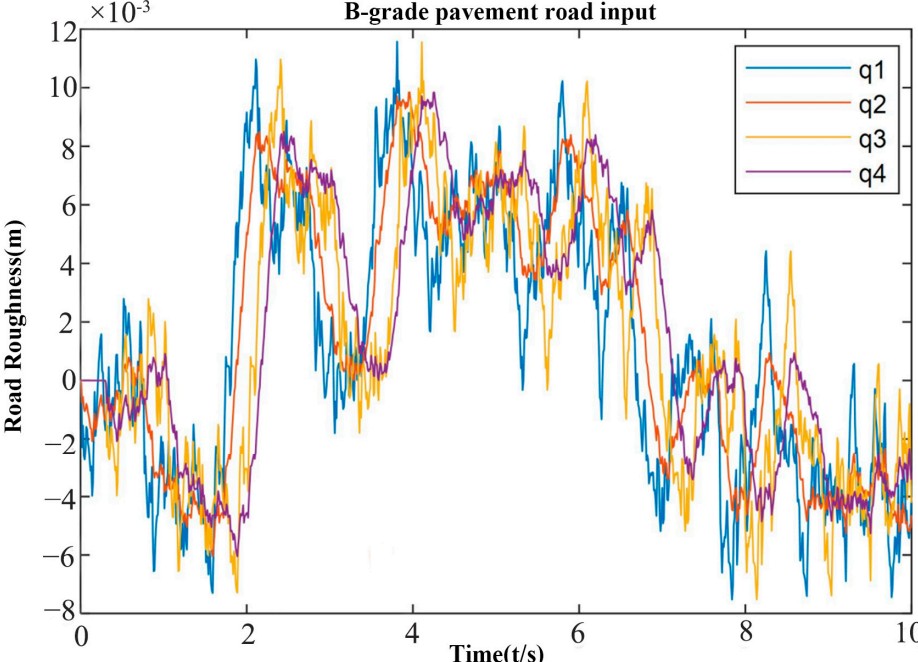

**Figure 5.** Four-wheel random-road input.

The value of the road input oscillates repeatedly around 0 and there is a time difference between front and rear axes, but their value is roughly the same.

## 3. Design of the Target-Distance-Recognition Method Based on the Binocular Camera

### 3.1. Speed-Bump-Detection Model Based on Deep Learning

In recent years, the emergence of convolutional neural networks (CNNs) has made great contributions to the development of target-detection technology, in which the one-stage target-detection algorithm YOLO (You Only Look Once) has been widely used because of its fast detection speed and strong model-generalization capability. Also, it is in the process of continuous updating and iteration [28]. In this paper, we adopted YOLOv4-tiny, a lightweight algorithm based on the improvement of the YOLOv4, that could meet the real-time accuracy requirements for the target detection of speed bumps during vehicle driving due to its simplified feature-extraction process and improved operation rate.

Then, the speed bump images were collected by real vehicles, and the photos of speed bumps with different shooting perspectives and different exposure degrees were selected to obtain effective data. To increase the sample size and improve the richness of the data, the dataset was expanded by the method of data augmentation. The images obtained are shown in Figure 6, in which a is the brightness enhancement, b signifies the contrast enhancement, c is the horizontal flip, and d is the random direction rotation. Finally, 4810 valid data were obtained, and the obtained image data were then target-labeled by the neural network dataset creation toolbox LabelImg to generate the VOC dataset, which produces a neural network readable dataset format.

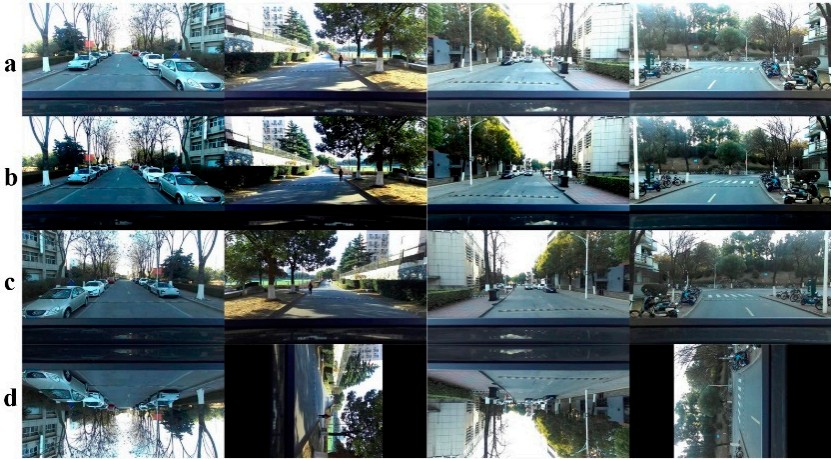

**Figure 6.** Data augmentation.

In the dataset, the training set was set to 2886 sheets, and both the test and validation sets were 962 sheets. To improve the training efficiency and training effect, migration training was performed using the existing pretraining weights. In the freeze training phase, the network parameters were fine-tuned without changing the parameters of the backbone network. By then, the Batch_size = 8 and epoch = 50. In the thaw training phase, all the network parameters were adjusted, with the Batch_size = 2 and epoch = 2950. After 3000 training cycles, the model converged and the training effect reached the real-time accuracy requirements. The main performance indicators of the model can be seen in Table 1.

**Table 1.** Model training performance index.

|  | Precision | F1 Score | Recall | mAP | AP |
|---|---|---|---|---|---|
| IoU = 0.5 | 98.57% | 0.99 | 98.57% | 98.53% | 98.53% |

The IOU is the intersection and concurrency ratio of the bounding box predicted by the model and the true detection box. Precision is the accuracy of model. Recall represents the proportion of positive classes predicted by the model that are indeed positive classes.

The F1 score is the reconciled average of precision and recall. The mAP is the mean average precision, and AP is the average precision.

### 3.2. Algorithm for Detecting the Start Point of a Circular Curve at the Lane Line

#### 3.2.1. Lane Line Detection

The recognition of the lane-line circle curve was based on lane line detection, which was achieved by preprocessing, lane line positioning, and lane line fitting.

The purpose of preprocessing was to remove noise and information that was not related to the detection target. After the original image optimization, ROI extraction, inverse perspective transformation, and binary image edge detection, the processed lane line image can be obtained, and the results are found in Figure 7.

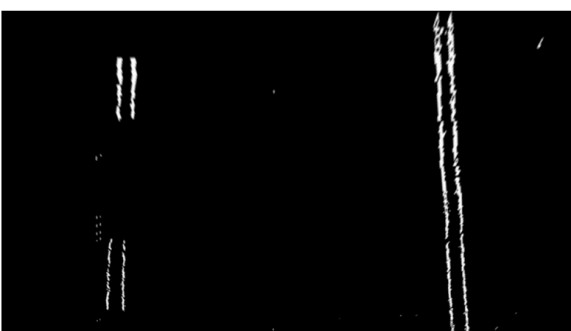

**Figure 7.** Lane line edge extraction.

#### 3.2.2. Lane Line Positioning

The white pixels in the bird's-eye view represent the lane outline and the black pixels denote the background information; therefore, it is possible to locate the lane line by obtaining the position of the white pixels. A sliding window is typically used to calculate the number of white pixels in each column of the pixel coordinate system, and the result is shown in Figure 8.

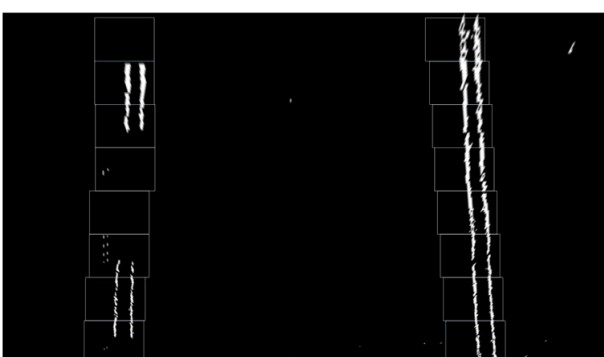

**Figure 8.** Sliding window positioning method.

To complete the sliding window lane line positioning, we need to determine the starting point of the lane line. The bird's-eye view from the bottom to the top represents a gradual distance from the camera. So, the sliding window is used to count the number of white pixel blocks in the window starting from the bottom, and the starting point of the lane line is determined based on the peaks of the number. Then, a new sliding window is drawn centered on the starting point towards the top, and the number of white pixels in the window is counted. The peak value is used as the new starting point to make a new sliding window. Then, the process is repeated to determine the direction of the lane line. The algorithm flow is shown in Figure 9.

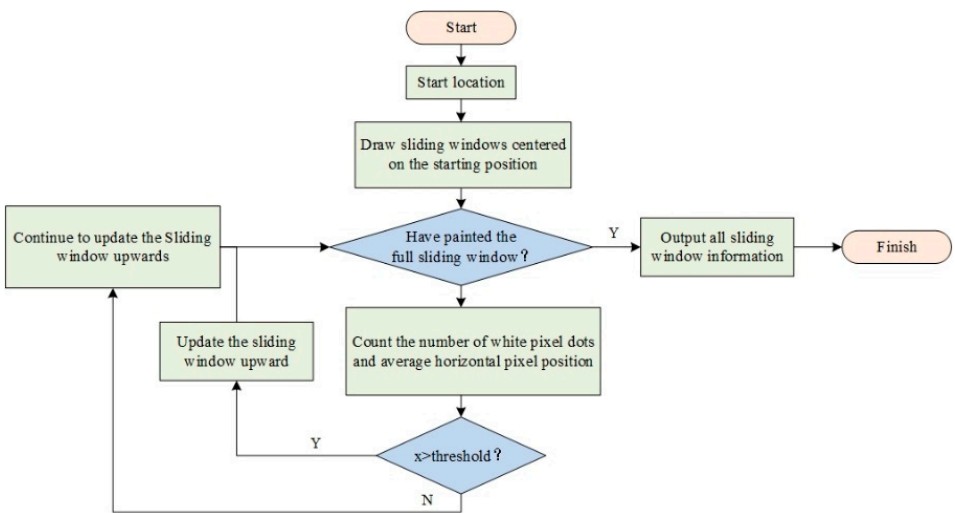

**Figure 9.** Sliding window positioning algorithm process.

### 3.2.3. Lane Line Fitting

The fitting algorithms applicable to the curved lane were mainly random sampling, the consistent RANSAC method, the Bessel curve fitting method, and the polynomial fitting method based on least squares. Given that the curvature of the lane lines in the curves within the urban roads would not change quickly, the lane line could be approximated as parabolic. The least-squares method was used in this paper to fit the lane lines [29,30].

The results obtained by fitting the lane lines based on this method are shown in Figure 10:

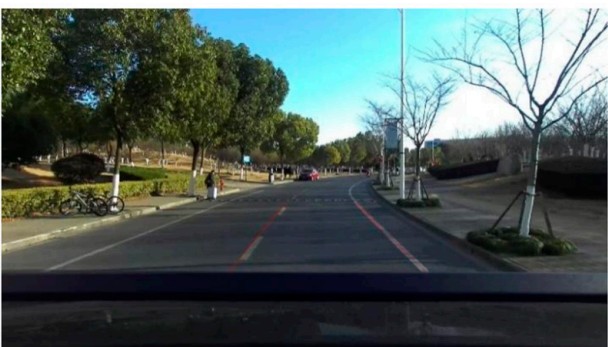

**Figure 10.** Lane line fitting.

### 3.2.4. Kalman Filtering

Since the least-squares method fitted the lane lines by traversing all the data only once, this method was less computationally intensive, but led to poor interference immunity. This method extracted lane lines based on fixed colors, and could be easily affected by the environment; as a consequence, errors and fitting jumps would be induced. The application of the Kalman filtering algorithm reduced this effect and fitted the lane lines more accurately [31]. Figure 11 is a comparison chart of lane-line-detection effect, and it can be seen that the application of Kalman filtering could improve the robustness of lane-line detection.

### 3.2.5. Lane-Line Curve Start-Point Location Detection

The minimum radius of the curvature of a circular curve at the design speed of an urban road section was found to be 100–200 m by checking the "Highway Route Design Specification" (JTG D20-2006). The recommended value was five to eight times the minimum value. Therefore, whether the curvature of the circular curve was greater than 1000 m

was chosen as the basis for judging whether to enter the curve. According to the radius of curvature calculation formula, it can be obtained:

$$R = \frac{1}{k} \tag{14}$$

Then, the curvature needs to satisfy:

$$k < \frac{1}{1000} \tag{15}$$

where $R$ is the radius of curvature and $k$ is the curvature:

$$k = \frac{y''}{\left[1 + (y')^2\right]^{\frac{3}{2}}} \tag{16}$$

Assuming that the position of the starting point of the circular curve under the image coordinate system is $(x_s, y_s)$, the expression of the fitted curve of the lane line obtained according to the Kalman filtering is:

$$y = a_f x^2 + b_f x + c_f \tag{17}$$

Equations (15) and (16) are joined to provide:

$$\frac{y''}{\left[1 + (y')^2\right]^{\frac{3}{2}}} < \frac{1}{1000} \tag{18}$$

The derivation of Equation (17) gives:

$$y'' = 2a_f \tag{19}$$

$$y' = 2a_f x + b_f \tag{20}$$

Bringing $x = x_s$ into Equations (19) and (20), and taking the result into Equation (18), gives:

$$\frac{2a_f}{\left[1 + \left(2a_f x_s + b_f\right)^2\right]^{\frac{3}{2}}} < \frac{1}{1000} \tag{21}$$

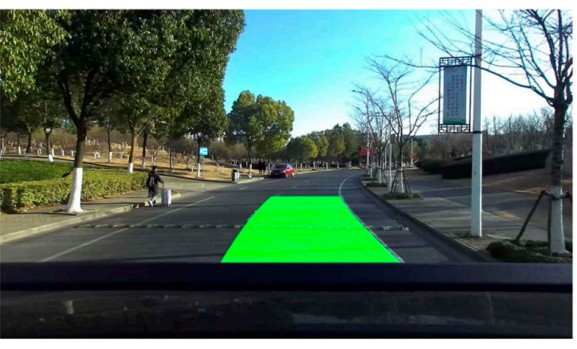 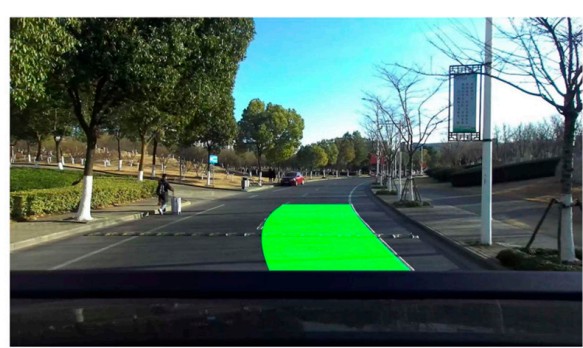

**Figure 11.** Comparison of lane-line-detection effects.

From the above equation, we could identify the range of $x_s$. The greatest value was selected and bought into the fitting curve to complete the positioning of $(x_s, y_s)$. The result is shown in Figure 12.

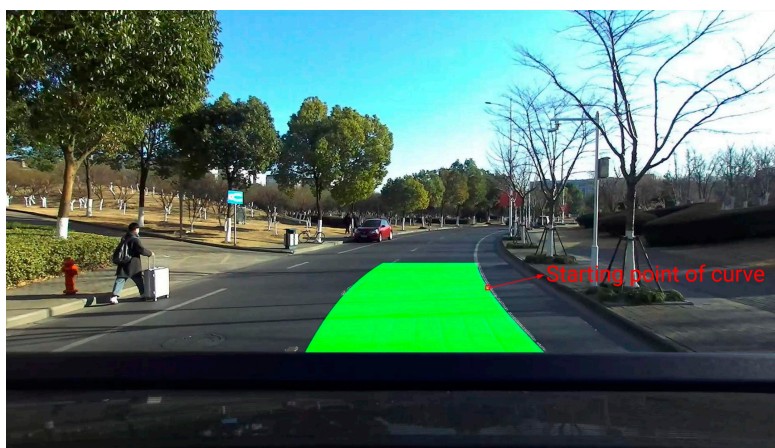

**Figure 12.** Lane-line circle-curve starting-point detection.

### 3.3. Target-Distance-Recognition Algorithm

In extracting the speed bump distance, the center position of the target-detection frame was selected as the distance-recognition extraction point because the target-detection frame could contain the speed bump, which was approximately rectangular in shape. If the speed bump was detected, the camera could get the upper-left vertex position $(u_{lu}, v_{lu})$ and lower-right vertex position $(u_{rl}, v_{rl})$ of the detection frame, and the principle is shown in Figure 13. The three coordinate systems from left to right are the camera coordinate system, the pixel coordinate system and the world coordinate system. The center position of the detection frame $(u_c, v_c)$ is derived from Equations (22) and (23):

$$u_c = \frac{(u_{rl} - u_{lu})}{2} + u_{lu} \tag{22}$$

$$v_c = \frac{(v_{rl} - v_{lu})}{2} + v_{lu} \tag{23}$$

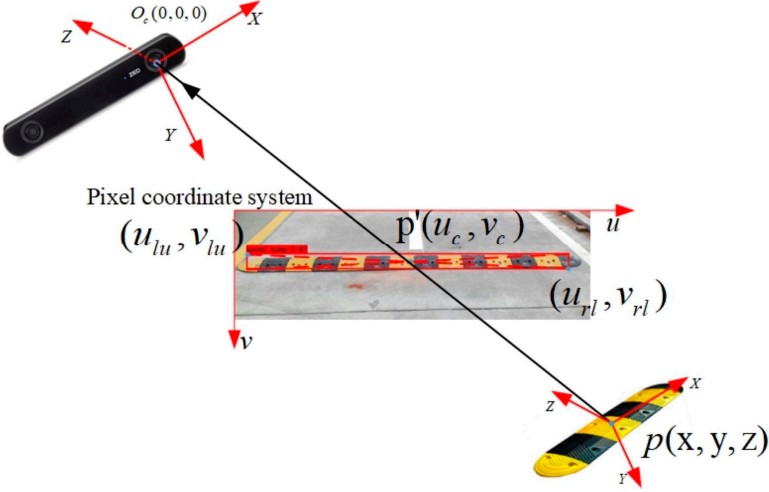

**Figure 13.** Coordinate relationship of the center point of the detection frame.

After determining the position of the center of the detection frame, the distance between the target and the camera could be calculated according to the coordinate conversion relationship of the camera. Then, the distance d between the speed bump and the

center of the front wheel could be calculated according to the parameters of the camera. The calculation formula is shown in Equation (24).

$$d = \sqrt{d_t^2 - h^2} - l_{cw} \tag{24}$$

where *h* refers to the height of the camera from the horizontal road surface and $l_{cw}$ denotes the horizontal distance from the camera to the center of the front axis.

The camera used in this paper was a binocular camera with depth-perception function, which could provide 3D point cloud information of the subject. In extracting the distance of the starting point of the circle curve, since the pixel-coordinate-location identification method of the starting point of the road circle curve has been given in the previous paper, the distance of the starting point of the circle curve could be extracted from the 3D point cloud data of the camera under the condition of the known pixel coordinates. The workflow of this extraction method is shown in Figure 14. If the lane-line circular curve start point is detected, the pixel coordinates are obtained and the point cloud information is calculated. Then, the distance from the target is calculated based on the distance conversion. If the start point is not detected, the lane line is continued to be detected.

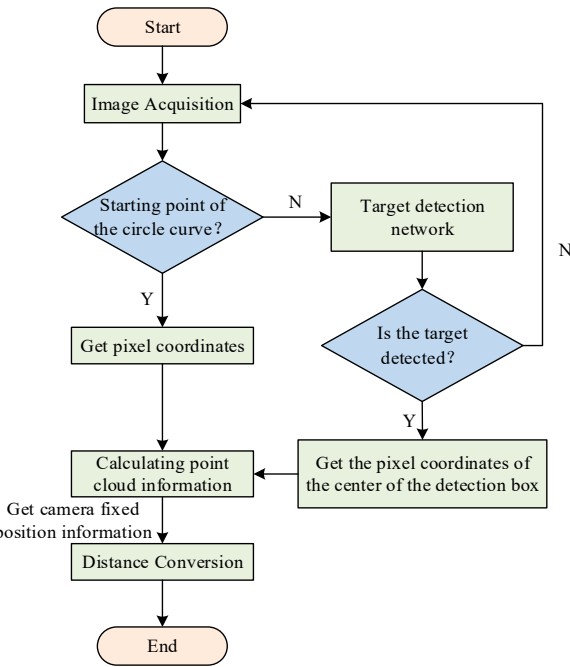

**Figure 14.** Target-distance extraction algorithm flow chart.

## 4. Design of the Intelligent Suspension Control System

The whole control system consisted of the road input, perception algorithm, control-mode-switching strategy, BP-PID controller, and force distributor. Firstly, the target was detected by the visual-perception system, and the time to reach the target was calculated according to the distance and driving speed. This was used to determine which control strategy the system switched to. Then, the difference between the desired output and the actual output of the control target was adopted as the input of the BP-PID controller in different control modes. The output of the control force was then obtained by self-tuning the PID parameters through the BP neural network. Subsequently, the output of control force was input to the force distributor to obtain the damping force on the four suspensions. Finally, the control current was calculated by the magnetorheological inverse model identification result, and the result could be input to the magnetorheological dampers to obtain the actual control force. The framework of the whole control system is shown in Figure 15:

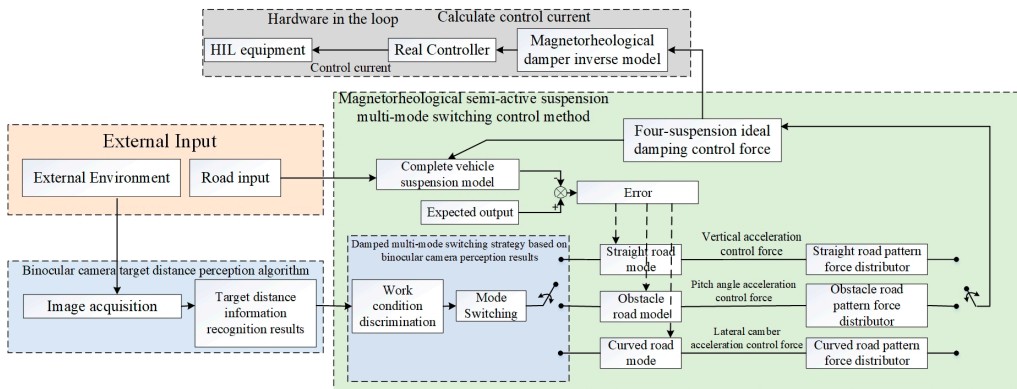

**Figure 15.** Semiactive suspension control system framework.

### 4.1. Mode Switching

The suspension control mode could be divided into the obstacle-road mode, straight-road mode, and curved-road mode according to different working conditions. The straight-road mode mainly controls the body dip acceleration to reduce the bumps caused by the road undulation, the obstacle-road mode mainly controls the body pitch acceleration to reduce the vehicle pitch-angle change due to the speed bump, and the curved-road mode mainly controls the body roll acceleration to reduce the rapid body tilt caused by the inertia of the spring mass when the vehicle enters the curve. At the same time, the priority of the curved-road mode was set to the highest to prevent the vehicle from being in a dangerous situation and causing injury to the passengers. The default control mode was the straight-road mode.

The working-condition identification method is shown in Figure 16. The vehicle started with $m = 0$ and entered the straight-road mode. Then, the vehicle determined whether to enter the curved-road mode. When $c = 1$, the first recognized time $t$ and the first distance $L_{t\_s}$ will be recorded. When $t \geq \frac{L_{t\_s}}{v}$, the vehicle enters the curved-road mode, otherwise continues to stay in the straight-road mode. Here, c equals 1 when traveling in a curve. When the vehicle is about to exit the curve, it will recognize the lane-line round curve start point for the last time. At this time, $t_e$ as well as $L_{t\_e}$ will be recorded. If $t_e \geq \frac{L_{t\_s}}{v}$, it means that the vehicle enters into the straight road and returns to the straight-road mode, otherwise it continues to stay in the curved-road mode.

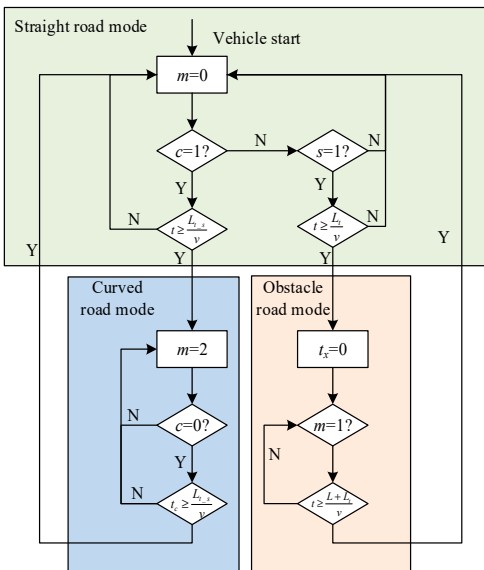

**Figure 16.** Condition identification method.

When the vehicle does not satisfy the judgement condition of the curve mode, a judgement will be made to determine whether the vehicle satisfies the switching condition of the obstacle-road mode. If $s$ = 1, the first recognized time t and the target distance $L_t$ will be recorded. If $t \geq \frac{L_t}{v}$, it means that the vehicle is passing the speed bump and goes into the obstacle-road mode. At this time, $m$ = 1. The time required for vehicles to completely pass the speed bump is used as the judgment basis for the duration of the obstacle-road mode. If $t \geq \frac{L_t+L}{v}$, it means that the vehicle completely passes the speed bump and it returns to the straight-road mode. At this time, $m$ = 0.

### 4.2. BP-PID Controller Design

The controller consists of the BP neural network and the PID controller. In 1986, Rumelhart et al. first introduced the concept of the BP neural network, which is a multilayer feedforward network based on the error back-propagation algorithm for network training. It continuously adjusted the network weights through sample data training, so that the error value decreased and the actual output results were close to the desired value. The error back-propagation algorithm ensures that the BP neural network has good generalization and self-learning ability, so it is widely used in the design of nonlinear system controllers. The input of the BP neural network includes the desired input, actual output, and the error of the system. The output consists of the adjustment parameters $K_p$, $K_i$, and $K_d$ to realize the online adjustment of the PID parameters. As such, it can realize a fast response under the uncertain input conditions of the time-varying system. The framework of the controller is shown in Figure 17.

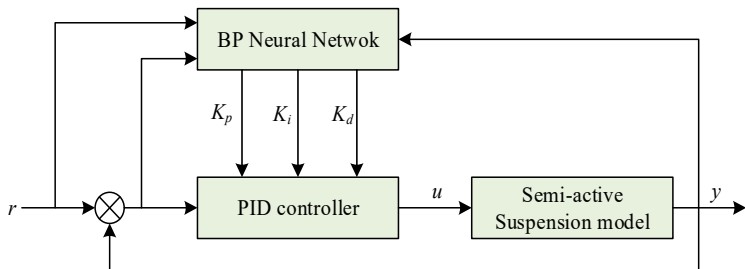

**Figure 17.** BP-PID controller.

The number of neurons m in the neural network was three. We selected the single-hidden-layer network, and the number of output neurons l was three. The sampling period $T_{bp}$ = 0.001 was designed as per the Simulink simulation step. The weights of each layer were initialized using random numbers from 0 to 1, and the Softmax function was used as the activation function.

In the case of the controller for the straight-ahead mode, the control objective of this mode was to keep the body smooth and reduce the body droop acceleration. Therefore, the body droop acceleration was adopted as the actual input of the controller, and the desired output was set as 0. The suspension control force was calculated by the neural network.

### 4.3. Parameter Optimization

Since the control forces obtained by the BP-PID controller were distributed to the four suspensions in the same direction and in equal amounts, the road inputs of the left and right wheels were not equal, and the effect of suppressing the body roll and pitch was not significantly improved. The next step was to design a force distributor to achieve a coordinated distribution of the damping forces.

4.3.1. Improvement of the Salp Swarm Algorithm

The salp swarm algorithm, proposed by Mirjalili et al. in 2017, was inspired by the idea of a "leader-follower chain" moving together to forage for food. The leader searches globally over a large area, while the followers follow the previous leader to explore locally

and eventually lead the group to food. The method improves global exploration and local exploitation, and reduces the number of local optima [32].

The flow of salp swarm algorithm is shown in Figure 18:

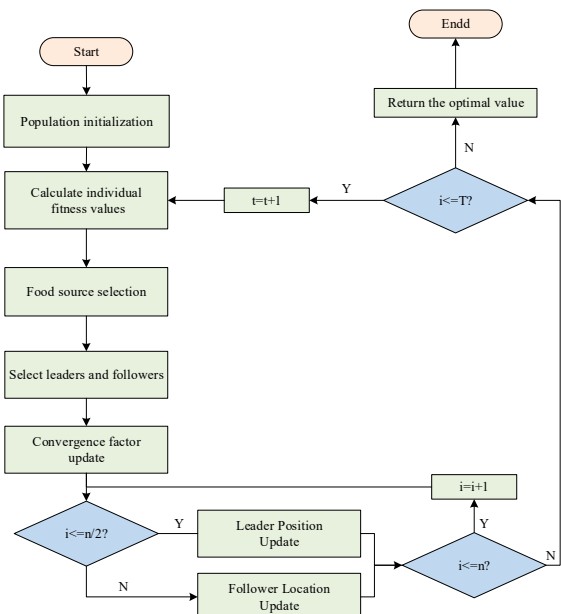

**Figure 18.** Flow of the salp swarm algorithm.

The leader position is updated according to Equation (26) and the follower position is updated according to Equation (27):

$$X_d^1 = \begin{cases} F_d + c_1(ub - lb)c_2 + lb, c_3 \geq 0.5 \\ F_d - c_1(ub - lb)c_2 + lb, c_3 < 0.5 \end{cases} \tag{25}$$

$$X_d^{i'} = \frac{X_d^i + X_d^{i-1}}{2} \tag{26}$$

$$c_1 = 2e^{-\left(\frac{4l}{L}\right)^2} \tag{27}$$

where $X_d$ and $F_d$ represent the $d$th dimensional leader position and food position; $u_b$ and $l_b$ denote the upper and lower bounds; c refers to the convergence factor, which is updated by Equation (28); $l$ is the number of current iterations and $L$ is the maximum number of iterations; $c_2$ and $c_3$ are random parameters in the range [0, 1].

The initialization of the bottle sheath group affects the computational speed as well as the accuracy of the algorithm. The use of the Tent Map to randomly generate a uniformly distributed initialization of the bottle sheath groups in the search space can be beneficial in improving the algorithm to find the optimal bottle sheath group, which can be expressed by Equation (29):

$$z_{j+1}^i = \begin{cases} \frac{z_j^i}{u}, 0 \leq z_j^i \leq u \\ \frac{1-z_j^i}{1-u}, 0 \leq z_j^i \leq 1 \end{cases} \tag{28}$$

where $i$ = 1, 2, 3, … N is the number of populations, $j$ signifies the number of current iterations, and u is the chaos control parameter. Thus, the Tent is applied to initialize the salp swarm as:

$$X_{D*N} = z_j^i(ub_{D \times N} - lb_{D \times N}) + lb_{D \times N} \tag{29}$$

In addition, the position of the food source moves all the time when the salp swarm is actually searching for food. So, the "crazy concept" is introduced to model this phe-

nomenon, and by introducing a madness operator in the leader position update equation to avoid the population from falling into local optimal solutions. The leader update equation is shown in Equation (31):

$$X_d^1 = \begin{cases} F_d + P(c_4)sign(c_4)x_{cr} + c_1(ub - lb)c_2 + lb, c_3 \geq 0.5 \\ F_d + P(c_4)\text{sign}(c_4)x_{cr} - c_1(ub - lb)c_2 + lb, c_3 < 0.5 \end{cases} \tag{30}$$

$$P(c_4) = \begin{cases} 1, c_4 \leq P_{cr} \\ 0, c_4 > P_{cr} \end{cases} \tag{31}$$

$$sign(c_4) = \begin{cases} -1, c_4 \geq 0.5 \\ 1, c_4 < 0.5 \end{cases} \tag{32}$$

where $c_4$ denotes a random number of [0, 1] conforming to a uniform distribution and $x_{cr}$ refers to a smaller constant; $P(c_4)$ is taken according to Equation (32), where $P_{cr}$ is the crazy probability; $sign(c_4)$ is taken according to Equation (33).

The dependence of followers on the leader depends on the inertia weight. A larger weight enhances the global search ability of the salp swarm and a smaller weight helps to achieve local exploitation. To balance the global search and local exploitation ability, linearly decreasing weights $w(t)$ were introduced, and the new follower update formula can be seen as follows:

$$X_d^{i'} = \frac{X_d^i + w(t)X_d^{i-1}}{2} \tag{33}$$

$$w(t) = \frac{w_s(w_s - w_e)(L - l)}{L} \tag{34}$$

where $w_s$ is the initial weight, $w_e$ represents the maximum number of iterations weight, $L$ is the maximum number of iterations, and $l$ refers to the current number of iterations.

### 4.3.2. Force Distributor Design

The objective functions to be optimized in the three modes can be constructed by the improved salp swarm algorithm, and the coordination weights of the damping forces in the different modes can be obtained. Taking the straight-road mode as an example, the forces assigned to the four dampers are:

$$f_i = d_{i_z} f_{bp_{pid}} \tag{35}$$

where $d_{i_z}$ is the damping force coordination weight.

Due to the different units and orders of magnitude of the vehicle acceleration and dynamic tire load, it is necessary to perform normalization. Taking the root mean square value of the corresponding performance index of the passive suspension as a reference, the subperformance function could be defined as follows:

$$p_\delta = \frac{RMS(\delta_{sa})}{RMS(\delta_{pas})} \tag{36}$$

where $RMS(\delta_{sa})$ is the root mean square value of the semiactive suspension performance index; $RMS(\delta_{pas})$ is the root mean square value of passive suspension; $\delta$ represents the body droop acceleration, body pitch-angle speed, body roll-angle acceleration, and tire dynamic load, respectively, while its subscript pas represents the passive suspension and *sa* represents the semiactive suspension.

The integrated optimization objective function is the sum of the subobjective functions, and the integrated optimization objective functions for the straight-road, obstacle-road, and curved-road mode are as follows:

$$P_{straight} = p_{\ddot{z}} + p_{F_{d1_{sa}}} + p_{F_{d2_{sa}}} + p_{F_{d3_{sa}}} + p_{F_{d4_{sa}}} \tag{37}$$

$$P_{obstacle} = p_{\ddot{\theta}} + p_{F_{d1_{sa}}} + p_{F_{d2_{sa}}} + p_{F_{d3_{sa}}} + p_{F_{d4_{sa}}} \tag{38}$$

$$P_{curved} = p_{\ddot{\phi}} + p_{F_{d1_{sa}}} + p_{F_{d2_{sa}}} + p_{F_{d3_{sa}}} + p_{F_{d4_{sa}}} \tag{39}$$

In addition, the suspension motion restraint and tire dynamic load restraint should be included to prevent the effects of prolonged abnormal operation on the suspension performance, as well as the vehicle. Suspension motion constraints and the dynamic tire loads can be expressed by Equations (40) and (41):

$$\frac{RMS(z_{us_n})}{[z_{us_n}]} \leq \frac{1}{3} \tag{40}$$

$$\frac{RMS(F_{dn})}{\left(m + \sum_{n=1}^{4} m_{un}\right)g} \leq \frac{1}{3} \tag{41}$$

where $z_{us_n}$ represents the suspension dynamic deflection and $[z_{us_n}]$ signifies the suspension dynamic travel.

Due to the addition of the constrained set of equations, it is necessary to remove the solutions outside the constraints by adding penalty terms to the integrated optimization objective function. The following equations can be used as the penalty terms:

$$p_{motion} = RMS(z_{us_n}) - \frac{1}{3}[z_{us_n}] \leq 0 \tag{42}$$

$$p_{force} = RMS(F_{dn}) - \frac{1}{3}\left(m + \sum_{n=1}^{4} m_{un}\right)g \leq 0 \tag{43}$$

Therefore, the integrated optimization objective function for the straight-road model is changed as follows:

$$P_{daily} = p_{\ddot{z}} + p_{F_{d1_{sa}}} + p_{F_{d2_{sa}}} + p_{F_{d3_{sa}}} + p_{F_{d4_{sa}}} + w_p p \tag{44}$$

where $w_p$ is the penalty weight and the value is 0 when the penalty term holds, otherwise it is 1; $p$ refers to the penalty factor and usually takes a larger value to exclude the unconditional solution.

The length of a salp was 1–10 cm and the median length was 5 cm, so the population number chain of the salp swarm did not exceed 300, and the number of populations was set to (50,4) considering the operation speed. The maximum number of iterations was 200, the inertia weight $w_s = 0.9$, $w_c = 0.4$, the madness probability $C_{cr} = 0.3$, and $x_{cr} = 0.0001$. When the value was less than three, or the maximum number of iterations was reached, the calculation was ended and the results were output.

The search range of the weight coefficients of the force distributor was set at $[-1, 1]$, and the obtained weight coefficients can be seen in Table 2. In the straight-road mode, the four forces are in the same direction. Under the obstacle-road mode, the rear axle forces are in the opposite direction to the front axle. In the curved-road mode, the left and right suspension forces are in opposite directions.

**Table 2.** Force controller weighting factor.

| Control Mode | Left Front | Right Front | Left Rear | Right Rear |
|---|---|---|---|---|
| Straight Road Mode | 0.951 | 0.986 | 0.933 | 0.915 |
| Obstacle Road Mode | 0.986 | 0.978 | −0.948 | −0.954 |
| Curved Road Mode | −0.972 | 0.961 | −0.981 | 0.986 |

### 4.4. Simulation Results and Analysis

In this paper, the joint simulation was realized by Python and MATLAB/Simulink. The perception algorithm was implemented based on Python3.8, which was the control algorithm. The whole vehicle and road-input model was implemented by Simulink2021a. Python passed to MATLAB2021a the lane-line circle-curve start-point result c, the speed-bump target-recognition result $s$, the time $t$, the target-object distance $L_t$, and other parameters. MATLAB obtained the control-mode-switching parameters m through Simulink with Python's parameters. The flow is shown in Figure 19.

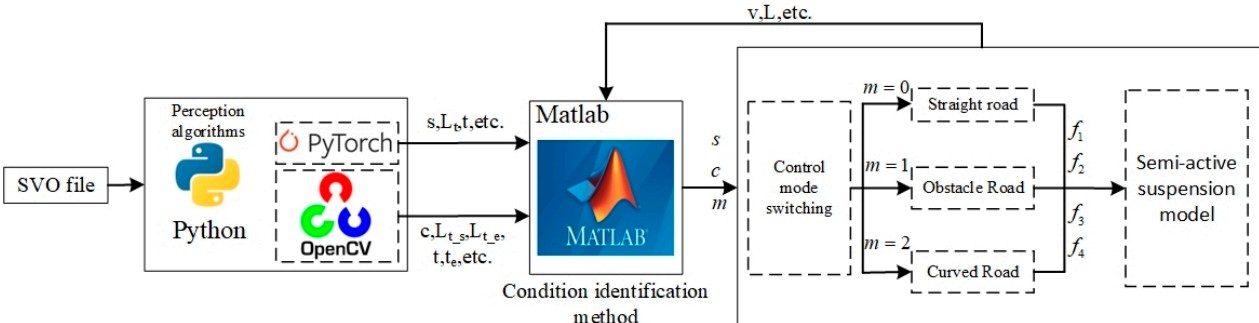

**Figure 19.** Flow of joint simulation.

The vehicle driving speed was stabilized at 10 m/s, and the total length of the test section was 250 m. After the measurement, the test section entered the curve at the position of 50 m, drove out of the curve after 110 m in the curve, and then had a speed bump at 70 m in the straight road. Figure 20 shows the satellite map of the collected road section and the result of the target-feature labeling.

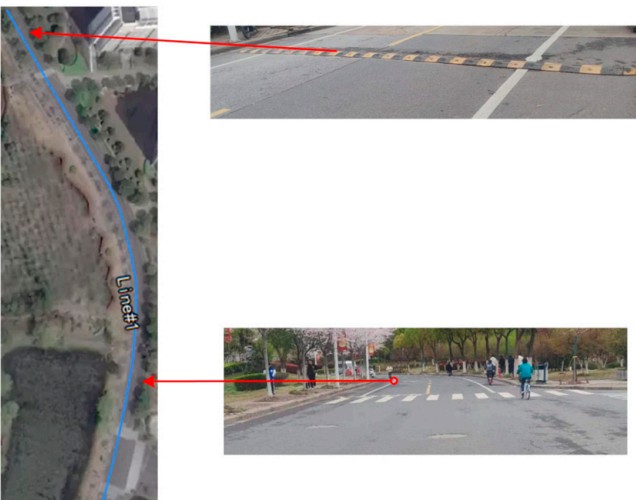

**Figure 20.** Image-acquisition section satellite map and feature mark.

The results of the joint simulation control mode switching are shown in Figure 21. The simulation switching curve was 0.009 s earlier than the theoretical switching curve into the straight-road mode, when the vehicle got out of the curve. The simulation test under the semiactive suspension control mode switched to the obstacle-road mode and was 0.008 s earlier than the theoretical switching curve when the vehicle was about to pass the speed bump target. After the vehicle passed the target of the speed bump, the simulated switching curve of the control method back to the straight-road mode was ahead of the theoretical switching curve by 0.007 s. The reason may be that the vehicle driving speed was slightly higher than 10 m/s, and the theoretical switching curve was calculated as 10 m/s. How-

ever, the difference between the simulation and theoretical model was only on the order of milliseconds, which would not affect the switching strategy efficiency.

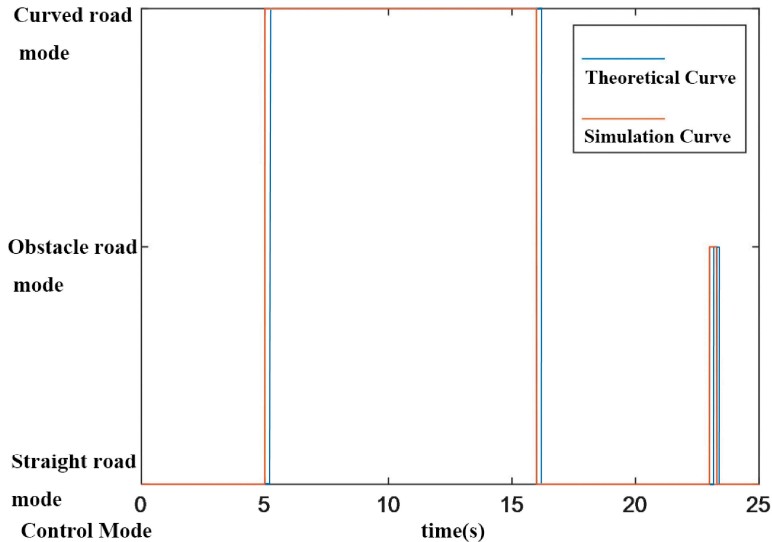

**Figure 21.** Control-mode-switching process.

The joint simulation of the suspension control effect is shown in Figure 22, and the results obtained are shown in Table 3:

**Table 3.** Decreased values of the semiactive suspension performance indicators.

| | Droop Acceleration | Roll-Angle Acceleration | Pitch-Angle Acceleration | Dynamic Deflection of Left Front | Dynamic Deflection of Right Rear | Dynamic Tire Load of Left Front | Dynamic Tire Load of Right Rear |
|---|---|---|---|---|---|---|---|
| Decrease percentage | 20.18% | 10.34% | 14.36% | 2.3% | 1.2% | 0.13% | 1.2% |

As can be seen, compared with the passive suspension, the semiactive suspension significantly improved the droop acceleration, and there was an over 10 percent improvement in the pitch and roll acceleration compared to the passive suspension. Meanwhile, the suspension dynamic deflection and the wheel dynamic load performance achieved a small improvement.

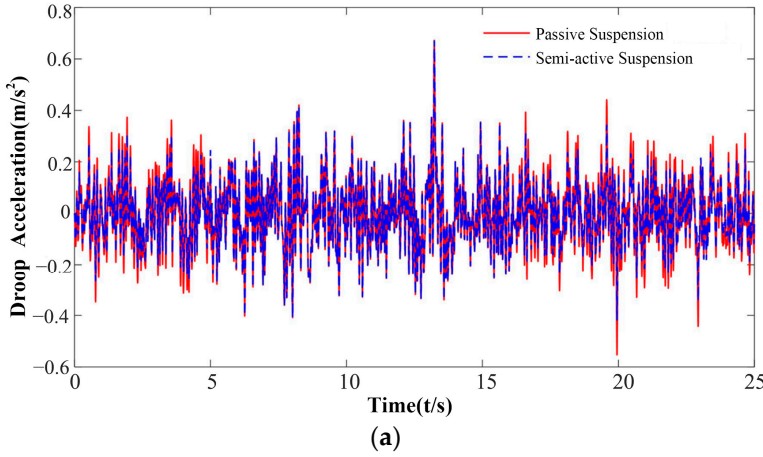

(a)

**Figure 22.** *Cont.*

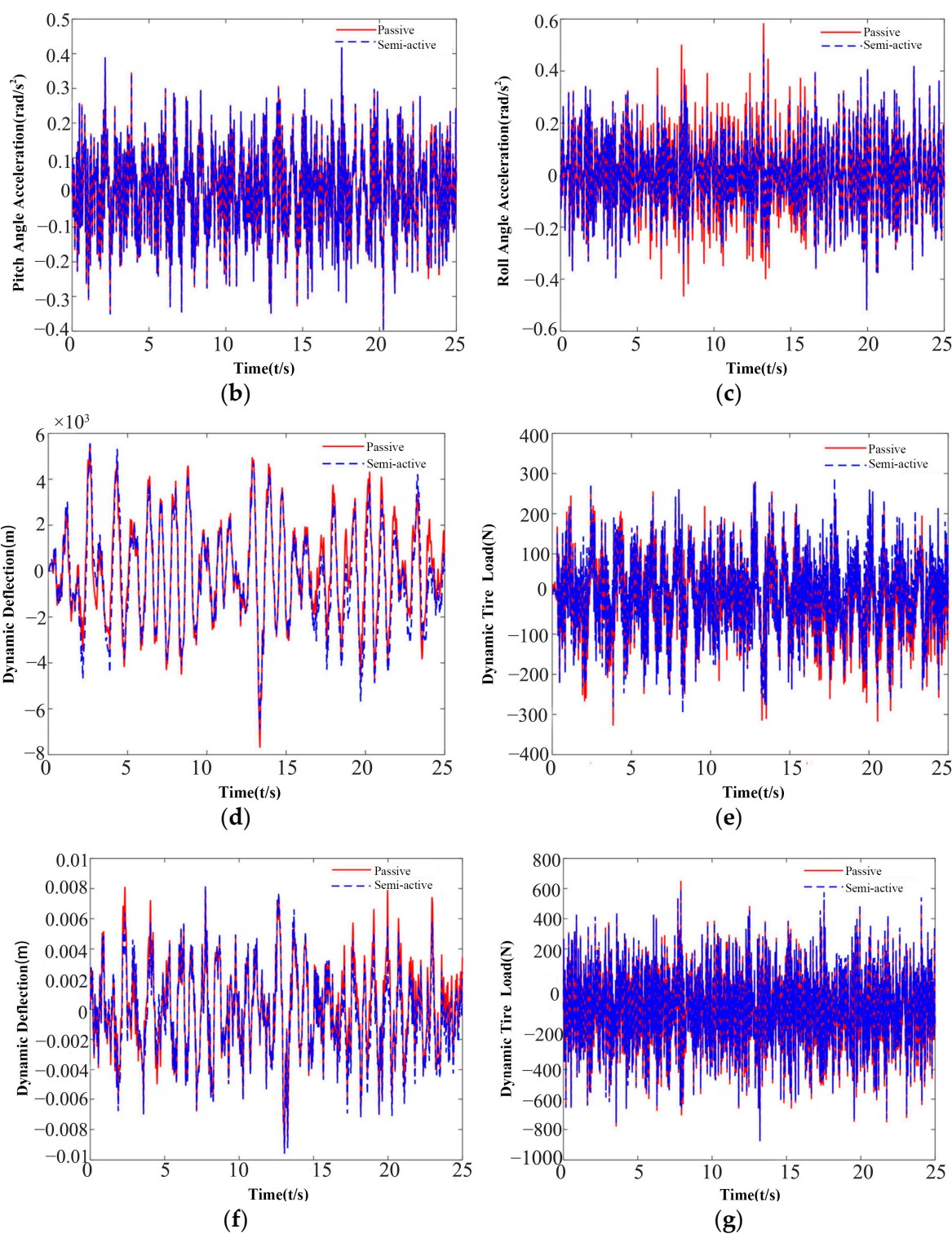

**Figure 22.** Simulation effect: (**a**) Comparison of the droop acceleration; (**b**) Comparison of the body pitch-angle acceleration; (**c**) Comparison of the body roll-angle acceleration; (**d**) Dynamic deflection of the right rear suspension; (**e**) Dynamic tire load of the right rear suspension; (**f**) Dynamic deflection of the left front suspension; (**g**) Dynamic tire load of the left front suspension.

## 5. HIL Experiment

The hardware-in-the-loop experimental test environment constructed in this paper consisted of the D2P (development to prototype), the host computer software (LABCAR-

OPERATOR V5.0) and the LABCAR test cabinet(from Vehinfo, Shanghai, China) [33]. The D2P connects and interacts with the host computer through the CAN bus and provides abundant I/O interfaces. The host computer was equipped with MATLAB, LCO, and other software, and the test interface for the host computer could be constructed to detect variables. LABCAR provided the I/O boards and CAN communication boards, which could test the operation effect, transmit the required signals to the host computer, and transmit the simulated signals during the simulation operation. The flow of the test is shown in Figure 23.

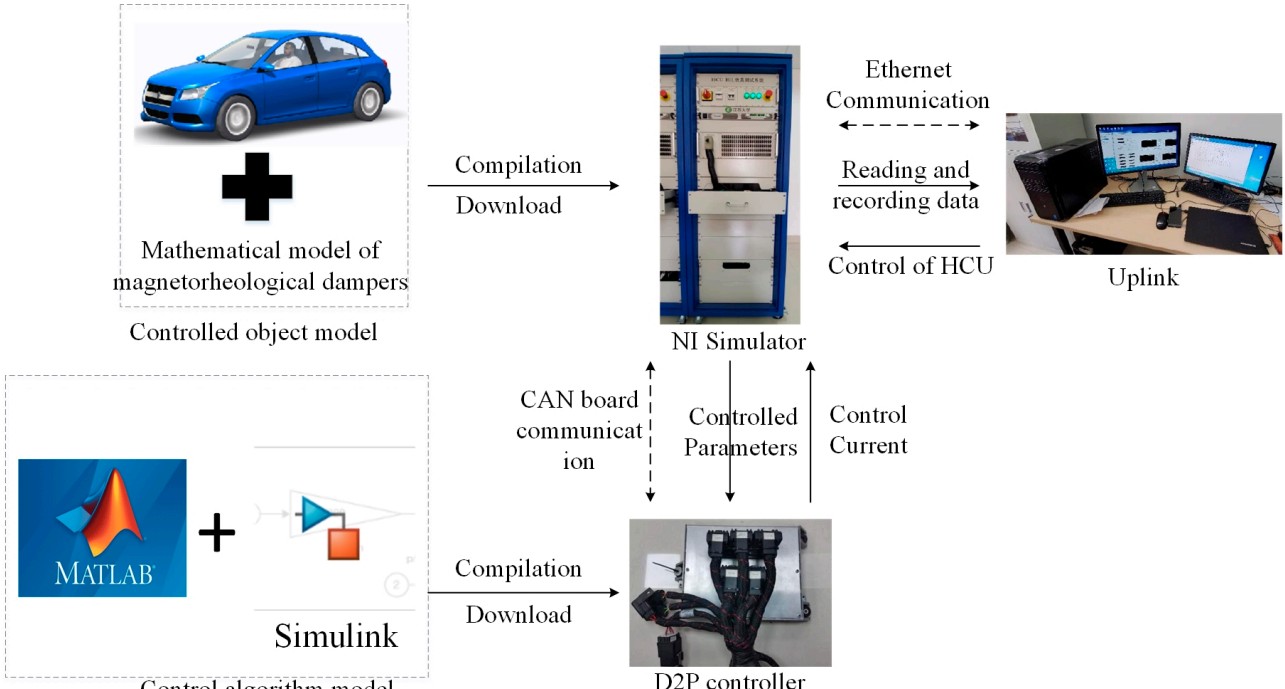

**Figure 23.** Flow of the HIL.

The hardware-in-the-loop test results are shown in Figure 24, the results obtained are shown in Table 4. And the analysis indicated that the root mean square value of body roll acceleration are more significantly improved compared to the pitch and droop acceleration. And the multi-mode control method improves the performance of suspension and tire compared to the passive suspension and PID control methods.

**Table 4.** Decreased values of the semiactive suspension performance indicators in the HIL.

| Decrease Percentage | Droop Acceleration | Roll-Angle Acceleration | Pitch-Angle Acceleration | Dynamic Deflection of Left Front | Dynamic Deflection of Right Rear | Dynamic Tire Load of Left Front | Dynamic Tire Load of Right Rear |
|---|---|---|---|---|---|---|---|
| Compared with PID | 1.2% | 2.4% | 10.12% | 3.22% | 2.75% | 0.56% | 0.92% |
| Compared with Passive | 3.79% | 4.45% | 22.27% | 5.62% | 6.62% | 1.85% | 2.86% |

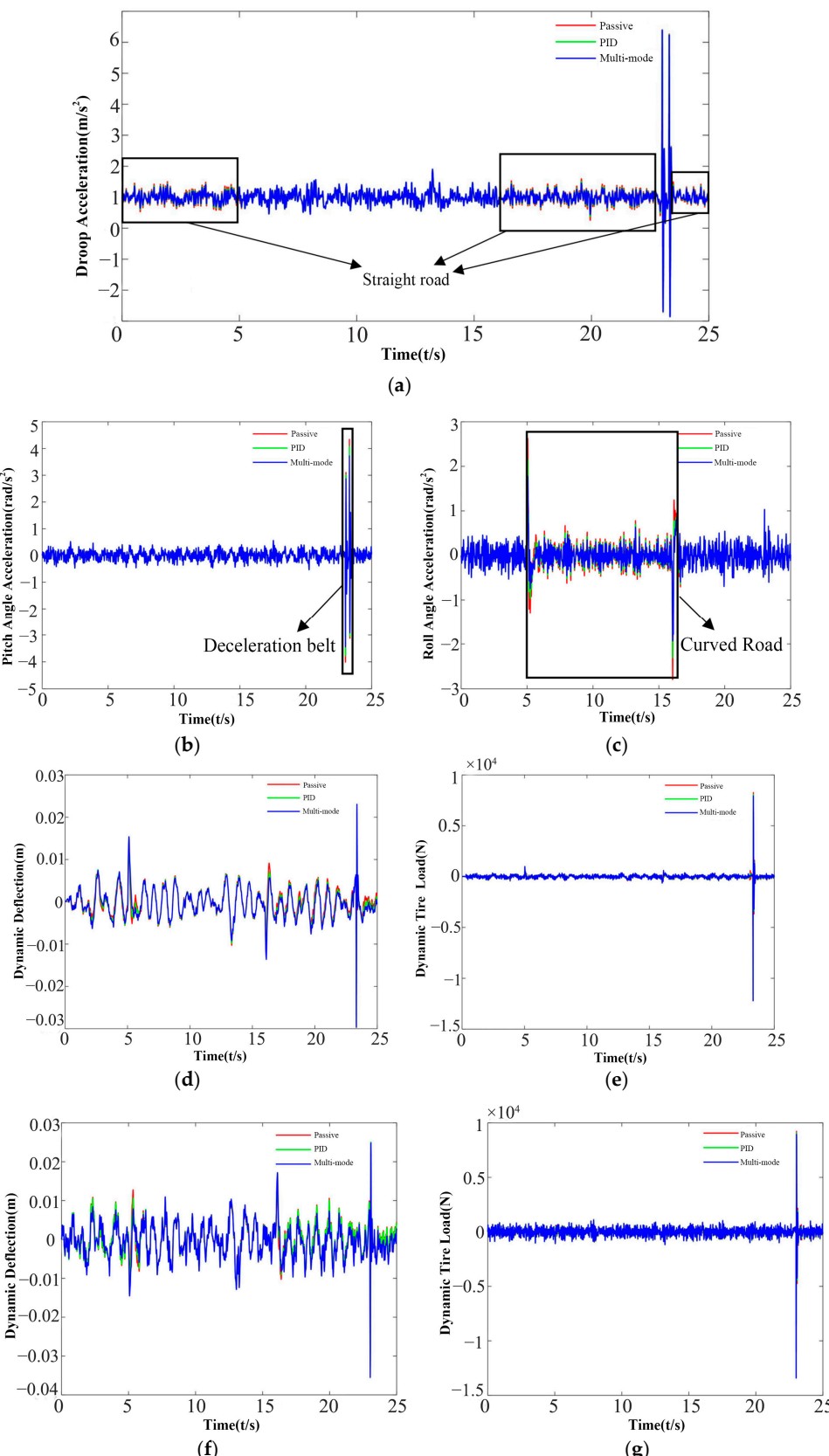

**Figure 24.** HIL test results: (**a**) Comparison of the droop acceleration; (**b**) Comparison of the body pitch-angle acceleration; (**c**) Comparison of the body roll-angle acceleration; (**d**) Dynamic deflection of the right rear suspension; (**e**) Dynamic tire load of the right rear suspension; (**f**) Dynamic deflection of the left front suspension; (**g**) Dynamic tire load of the left front suspension.

## 6. Conclusions

In this paper, a multimode control strategy for a semiactive suspension with a magnetorheological damper based on binocular camera target-distance recognition was designed. The aim was to realize the adaptive switching of semiactive suspension control methods under different road conditions so as to improve the smoothness, comfort, and handling stability of the vehicles. Although we made a little progress, there are still many of problems that need to be solved in the future due to the limitations of the experimental equipment, experimental conditions, and time:

(1) The effectiveness of the control method needs to be further verified from the perspective of a real vehicle. Due to the lack of a real vehicle equipped with a magnetorheological semiactive suspension, and the time relationship, this paper only carries out MIL and HIL experiments through Simulink and Carsim, and does not carry out real-vehicle tests.

(2) This paper makes less use of the function of the binocular camera. As a sensor, it can measure the 3D point cloud so that it has a powerful function that is not inferior to Laser Radar. By using the binocular camera to realize the real-time scanning of the front terrain, it can provide richer road-surface information for the suspension control algorithm, and realize the suspension control algorithm with a more excellent effect.

**Author Contributions:** Conceptualization, C.H. and K.L.; methodology, C.H.; software, K.L.; validation, C.H. and K.L.; formal analysis, C.H., K.L. and Q.X.; investigation, K.L.; resources, Y.D.; data curation, K.L.; writ-ing—original draft preparation, K.L.; writing—review and editing, C.H.; visualization, Q.X.; su-pervision, Q.X.; project administration, C.H.; funding acquisition, Y.D. All authors have read and agreed to the published version of the manuscript.

**Funding:** This work is supported by the State Key Laboratory of Vehicle Safety and Energy Saving of Tsinghua University, grant number [KFY2207], International Cooperation Fund of Jiangsu, grant number [BZ2022050], and General Program of China Postdoctoral Science Foundation, grant number [2021M691847].

**Data Availability Statement:** Data are contained within the article.

**Acknowledgments:** Thanks to the State Key Laboratory of Vehicle Safety and Energy Saving of Tsinghua University, and International Cooperation Fund of Jiangsu for the support of this paper.

**Conflicts of Interest:** The authors declare no conflict of interest.

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
