# Peer review of "Research on the Multimode Switching Control of Intelligent Suspension Based on Binocular Distance Recognition"

_wevj, doi:10.3390/wevj14120340_

Round 1
Reviewer 1 Report
Comments and Suggestions for Authors
attached

Reviewer 2 Report
Comments and Suggestions for Authors
1) At the end of section 1 "Introduction" briefly present all the following sections of the work.
2) The statement in lines 86-88 requires citation: "The seven-degree-of-freedom whole-vehicle dynamics model is shown in Fig. 1. The model is more consistent with the actual vehicle dynamics, so this paper chooses to use this model to develop the control algorithm."
3) Briefly present the dynamic model proposed in Figure 1 before it or immediately after the figure.
4) Absolutely all parameters used in equations 1-11 must be defined before their first appearance in the text or equation, including their measurement units.
5) Increase the resolution of Figure 2.
6) Discuss in more detail the results presented in the diagrams in Figures 3-4, immediately after each figure.
7) Write a few sentences about the elements of Table 1, immediately after the table.
8) Try to increase the resolution of Figures 6 and 7.
9) Figure 8 requires a discussion immediately after it.
10) Discuss the algorithm in Figure 13 immediately after the figure. It would be good to refer to Figure 13 before it instead of referring to the following figure.
11) Discuss the elements in Tables 2 and 3 immediately after their appearance in the paper.
12) Figure 20 requires a higher resolution.
13) Section 4 of the work called "HIL Experiment" should be marked with 5 (as section 5).
14) Increase the resolution of Figure 22.
15) Discuss in more detail the results presented in the diagrams in Figure 23.
16) The Conclusions section should be marked with 6 and developed a little to briefly include all the new aspects introduced in the work.
17) Insist on the following aspects when you highlight the main contributions of the work:
In order to optimize the control effect of the semi-active suspension under different working conditions, this paper completed the modeling of the dynamics of the magnetorheological semi-active suspension system and road intake.
Carrying out the design of the binocular camera detection algorithms to obtain the real-time distance of the target using the point cloud range function and obtaining the necessary parameters for the suspension control.
Completion of control mode switching rules and suspension controller design.
Verification of the effectiveness of the mode switching rules and control method through system simulation and hardware-in-the-loop test.
Comments on the Quality of English Language
Minor editing of the English language is required.
Round 2
Reviewer 1 Report
Comments and Suggestions for Authors
1. abstract: the expression shall be Neural Network PID controller (the fact that the NN is trained by Back Propagation can be said in the section describing the training of the NN; BP is the common method)
2. the passive dash pot elements c1, c2, c3, c4 (Fig.1) are still not explained in the text. They shall be explained. (Please notice that it does not make sense to have an additional passive damper besides the controllable MR damper.)
3. the legend of fig3 is not readable. What is the difference between fig. 3 and fig. 4. If Fig. 4 should show the model validation then authors need to plot the comparison between simulation and measured f-x-curves in seperate sub-plots.
Reviewer 2 Report
Comments and Suggestions for Authors
1) There are two Figures marked with 20.
2) After that, all the numbers of the Figures following 20 must be changed.
3) Conclusions must be improved and developed.
Minor editing of English language required
